# The MYB Transcription Factor *GmMYB78* Negatively Regulates *Phytophthora sojae* Resistance in Soybean

**DOI:** 10.3390/ijms25084247

**Published:** 2024-04-11

**Authors:** Hong Gao, Jia Ma, Yuxin Zhao, Chuanzhong Zhang, Ming Zhao, Shengfu He, Yan Sun, Xin Fang, Xiaoyu Chen, Kexin Ma, Yanjie Pang, Yachang Gu, Yaqun Dongye, Junjiang Wu, Pengfei Xu, Shuzhen Zhang

**Affiliations:** 1Soybean Research Institute of Northeast Agricultural University/Key Laboratory of Soybean Biology of Chinese Education Ministry, Harbin 150030, China; gaohong9@126.com (H.G.); 15504500812@163.com (J.M.); a15146568793@163.com (Y.Z.); zhangchuanzhong92@126.com (C.Z.); 13149511790@163.com (M.Z.); shengfuhe1996@163.com (S.H.); abcd905260884@163.com (Y.S.); fangxin0622@163.com (X.F.); chenx147687@163.com (X.C.); ma18106359538@outlook.com (K.M.); pyj0910@163.com (Y.P.); 18230381767@163.com (Y.G.); dongye15554737202@163.com (Y.D.); 2Soybean Research Institute of Heilongjiang Academy of Agricultural Sciences/Key Laboratory of Soybean Cultivation of Ministry of Agriculture, Harbin 150030, China; nkywujj@126.com

**Keywords:** GmMYB78, soybean, *Phytophthora sojae*, jasmonic acid, GmbHLH122, *GmbZIP25*

## Abstract

Phytophthora root rot is a devastating disease of soybean caused by *Phytophthora sojae*. However, the resistance mechanism is not yet clear. Our previous studies have shown that *GmAP2* enhances sensitivity to *P. sojae* in soybean, and *GmMYB78* is downregulated in the transcriptome analysis of *GmAP2*-overexpressing transgenic hairy roots. Here, *GmMYB78* was significantly induced by *P. sojae* in susceptible soybean, and the overexpressing of *GmMYB78* enhanced sensitivity to the pathogen, while silencing *GmMYB78* enhances resistance to *P. sojae*, indicating that *GmMYB78* is a negative regulator of *P. sojae*. Moreover, the jasmonic acid (JA) content and JA synthesis gene *GmAOS1* was highly upregulated in *GmMYB78*-silencing roots and highly downregulated in overexpressing ones, suggesting that *GmMYB78* could respond to *P. sojae* through the JA signaling pathway. Furthermore, the expression of several pathogenesis-related genes was significantly lower in *GmMYB78*-overexpressing roots and higher in *GmMYB78*-silencing ones. Additionally, we screened and identified the upstream regulator GmbHLH122 and downstream target gene *GmbZIP25* of GmMYB78. GmbHLH122 was highly induced by *P. sojae* and could inhibit *GmMYB78* expression in resistant soybean, and GmMYB78 was highly expressed to activate downstream target gene *GmbZIP25* transcription in susceptible soybean. In conclusion, our data reveal that *GmMYB78* triggers soybean sensitivity to *P. sojae* by inhibiting the JA signaling pathway and the expression of pathogenesis-related genes or through the effects of the GmbHLH122-GmMYB78-*GmbZIP25* cascade pathway.

## 1. Introduction

Phytophthora root and stem rot of soybean is a devastating worldwide soybean disease caused by the oomycete *Phytophthora sojae*, which can result in billions of dollars in economic losses globally each year on average [1,2]. Due to the complexity of the virulence changes in the *P. sojae* population, new pathogenic types are constantly emerging [3]. Therefore, it is necessary to study the genes that respond to *P. sojae* stress in order to better understand the response mechanism of soybeans under *P. sojae*-induced stress. These genes will provide useful information for genetic engineering and breeding.

The transcription factor (TF) is also known as the trans-acting factor [4]. Various plant transcription factors such as MYB, bHLH, AP2/ERF, WRKY, and NAC have been reported, with most of these transcription factors playing an important role in responding to the tolerance of plants to external environmental stresses [5,6,7,8,9]. Among many transcription factors, MYB is one of the most widely distributed and functionally strong members of the plant transcription factor family [10]. According to the number of MYB domains, MYB family is divided into four subfamilies: the 1R-MYB, R2R3-MYB, 3R-MYB, and 4R-MYB family [10]. The 1R-MYB family has only one MYB domain, which plays an important role in regulating plant transcription and maintaining the chromosome structure [11]. The R2R3-MYB family genes contain two conserved R2 and R3 repeat sequences in the MYB binding domain, and also include a regulatory domain (activating or inhibitory function) in the variable region at the C-terminus. This subfamily, with the most members and diverse functions among the four subfamilies, is widely involved in cell differentiation, secondary metabolism, and stress responses [12,13]. The 3R-MYB family genes have a conserved domain composed of R1, R2, and R3, mainly regulating cell differentiation and cell cycle control [14]. The 4R-MYB subfamily genes have a conserved domain composed of four R1/R2 repeat sequences, and the number of genes in this subfamily discovered in plants is currently very small [10]. 

Plant hormones are essential substances that regulate various physiological and biochemical reactions in plants, allowing normal life activities to proceed; The MYB gene expression in plants is closely related to plant hormones [15,16]. GhMYB18 can activate gene expression in salicylic acid (SA) and phenylpropane signaling pathways, promoting the synthesis of SA and flavonoid compounds [17]. Wheat TaMYB391 and TaMYB29 can promote gene expression in the SA signaling pathway, participating in the biosynthetic pathway of SA [18,19]. Research has found that AtMYB44 can interact with the Arabidopsis abscisic acid (ABA) receptor RCAR1/PYL9 and regulate the expression of the ABA response gene *RAB18*, participating in the ABA signaling pathway [20]. AcoMYB4 negatively regulates osmotic pressure by participating in ABA biosynthesis and signal transduction pathways [21]. FtMYB22 can interact with the ABA receptor proteins RCAR1/2 to form heterodimers and play a role in the ABA signaling pathway [22]. MYB7, as a negative regulator of ABA signaling, participates in seed germination by inhibiting the expression of *ABI5* (*ABA insensitive 5*) [23]. GmABAS1 directly binds to the promoter of *ABI5* to suppress its expression, thereby enhancing soybean sensitivity to ABA [24]. RhMYB108 can bind to the promoters of target genes *RhNAC053*, *RhNAC092*, and *RhSAG113*, participating in ethylene and jasmonic acid (JA) signaling pathways in roses (*Rosa hybrida*) [25]. The R2R3-MYB transcription factor GhMYB25 can interact with GhJAZ2 to regulate the JA signaling pathway [26]. Tea tree CsMYB46 and CsMYB105 can interact with CsJAZ3, CsJAZ10, and CsJAZ21 in the cell nucleus, thereby participating in the JA biosynthetic pathway [27]. 

MYB transcription factors play an important role in responding to fungal, bacterial, and oomycete infections [28,29,30,31]. For example, MdMYB73 can promote gene expression in the SA synthesis pathway, increase SA content, and enhance resistance to *Botryosphaeria dothidea* in apple [6]. TuMYB46L (*Triticum urartu*) can bind to the promoter region of the ethylene synthesis-related gene *TuACO3* to inhibit its expression, and the overexpression of *TuMYB46L* can reduce the ethylene content in wheat, leading to the increased susceptibility of wheat to *Blumeria graminis* f. sp. *tritici* [30]. The overexpression of *AtMYB44* in Arabidopsis downregulates the defense response against *Alternaria brassicola* [31]. Rice *myb* negatively regulates rice resistance to *Pyricularia grise* by participating in the JA signaling pathway [32]. *MdMYB30* enhances transgenic apple callus resistance to *Botryosphaeria dothidea* by regulating wax biosynthesis, and the overexpression of *MdMYB30* in Arabidopsis can enhance resistance to *Pseudomonas syringae* pv. tomato DC3000 [33]. Studies have shown that the overexpression of *CaMYB78* can increase chickpea resistance to *Fusarium solani* [34]. The MYB transcription factor CaPHL8 can enhance pepper resistance to *Ralstonia solanacearum* by directly or indirectly regulating the expression of defense-related genes [35]. PalbHLH1 and PalMYB90 can activate the expression of genes related to the flavonoid biosynthesis pathway, thereby increasing the accumulation of flavonoids in poplar and enhancing its resistance to fungal or bacterial infections [36]. The R2R3-MYB transcription factor CaMYB39 can induce the expression of genes involved in flavonoid biosynthesis to increase the accumulation of flavonoids, and the overexpression of *CaMYB39* can induce defense gene expression and enhance chickpea resistance to *Ascochyta rabiei* [37]. TaMYB391 and TaMYB29 can promote the expression of *PR* genes and SA signaling pathway genes, and then increase the accumulation of SA, thereby regulating the defense response to *Puccinia striiformis* f. sp. *tritici* [18]. Furthermore, knocking out *TaMYB4* weakens the resistance of wheat to *Puccinia striiformis* f. sp. *tritici* [38]. *VdMYB1* (*Vitis davidii*) enhances the resistance to *Erysiphe necator* by binding to the promoter of the key regulatory factor of flavonoid metabolism, *Stilbene Synthase* (STS), and activating *STS* expression [39]. *PnMYB2* (*Panax notoginseng*) positively regulates its resistance to *Fusarium solani* by modulating JA signaling, photosynthesis, and the expression of the disease-resistant gene [40].

Our previous study identified an AP2/ERF transcription factor, GmAP2 (Phytozome. Glyma.03G136100), which negatively regulates soybean resistance to *P. sojae* [41]. A downregulated MYB transcription factor named GmMYB78 was identified through a transcriptome analysis of *GmAP2*-overexpressing transgenic hairy roots. In this study, *GmMYB78* (GenBank accession no. XM_003536811.5) was isolated from susceptible soybean cultivar ‘Dongnong 50’, and its involvement in the defense response against *P. sojae* was investigated. The overexpression and RNA interference analyses indicated that *GmMYB78* negatively regulates soybean resistance to *P. sojae*, possibly through the downregulation of pathogenesis-related gene expression, JA synthesis, and signal transduction pathways. In addition, it may also be through the GmbHLH122 transcriptional inhibition of *GmMYB78* and the GmMYB78 transcriptional activation of *GmbZIP25* that inhibits resistance to *P. sojae*.

## 2. Results 

### 2.1. GmMYB78 Is Induced by P. sojae in Susceptible Soybean Cultivar and Localized in the Nucleus

The full-length cDNA sequence of *GmMYB78* (GenBank accession no. XM_003536811.5) was obtained from the total RNA of the soybean cultivar ‘Dongnong 50’. The sequence analysis showed that the full-length cDNA of the *GmMYB78* gene was 1160 bp, containing a 711 bp open reading frame (ORF), encoding 236 amino acids, with two MYB domains, belonging to the R2R3 MYB family transcription factor (Appendix A). Phylogenetic analysis and multiple sequence alignment demonstrated that GmMYB78 shares 60.98–99.58% identity in overall amino acid sequence with its homologous genes, including GsMYB78-like (XP_028183329.1), GsMyb-related protein 305 (KHN00328.1), SsMYB108 (TKY52432.1), GmMYB111 (ABH02873.1), CcMYB108 (XP_020228609.1), GbJAMYB-like (XP_061373619.1), CcMYB21 (KYP56143.1), ApJAMYB-like (XP_027367890.1), TpMYB (PNY03497.1), MaMYB (QSD99703.1), VuMYB2 (XP_027934157.1), VuMYB2-like (XP_047159236.1), VaMYB2 (XP_017413875.1), TpMYB78-like (XP_045807756.1), AhMYB2 (XP_025618640.1), MtMYB (XP_003591357.1), VuMYB (QCE07124.1), SsMYB2 (MED6186751.1), MtMYB61 (ABR28342.1), AdMYB2 (XP_015939424.1), CcMYB2 (XP_020228610.1), TrMYB (WJX19791.1), AiMYB78-like (XP_016177558.1), TrMYB108 (KAK2403414.1), VrMYB2-like (XP_014512333.1), SbMYB (KAJ1425627.1), QsMYB (KAJ7973106.1), GbMYB108-like (XP_061366235.1), LjMYB108-like (XP_057428076.1), PsMYB2-like (XP_050888452.1), TrMYB108 (KAK2456963.1), CcMYB108 (RDX74227.1), CcMYB21 (KY33749.1), CcMYB2 (XP_020207548.1), GmMYB2 (XP_003520762.1), SsMYB108 (TKY61683.1), AhMYB108 (XP_025618639.1), RcMYB78 (XP_028768938.1), TwMYB78 (XP_019454456.1), EgMYB78 (XP_019432183.1), JcMYB108 (XP_012084138.1), PaMYB108 (XP_034932063.1), and PtMYB108 (XP_021279389.1) (Appendix A).

In order to analyze the expression profiles of *GmMYB78*, we used RT-qPCR to detect the transcription level of *GmMYB78* in the resistant soybean cultivar ‘Suinong 10’ and the susceptible soybean cultivar ‘Dongnong 50’. As shown in Figure 1A, the relative expression levels in roots, leaves, and cotyledons of the susceptible soybean cultivar ‘Dongnong 50’ were significantly higher than those of the highly resistant soybean cultivar ‘Suinong 10’ (** *p* < 0.01). We further analyzed whether *GmMYB78* was induced by *P. sojae*. The results showed that the expression of *GmMYB78* in ‘Dongnong 50’ increased rapidly after inoculation with *P. sojae*, and reached the peak at 9 h, and then decreased gradually. By contrast, the expression of *GmMYB78* in ‘Suinong 10’ was rapidly downregulated at 6 h post-inoculation (hpi) (Figure 1B). It was concluded that *GmMYB78* could be induced by *P. sojae*, and the expression level of *GmMYB78* was different between resistant and susceptible soybean cultivars. The expression level of *GmMYB78* in soybean ‘Dongnong 50’ was significantly higher than that in soybean ‘Suinong 10’, indicating that *GmMYB78* may be a negative regulator in response to *P. sojae* infection.

In addition, we assessed the subcellular localization of GmMYB78 in Arabidopsis protoplasts that co-transfected the *GmMYB78*-GFP construct or 35S:GFP control construct (Figure 1C) with the nuclear marker H2B-mCherry (encoding histone H2B fused to the red fluorescent protein mCherry). As shown in Figure 1D, we detected GFP fluorescence throughout the cells expressing the 35S:GFP control plasmid. By contrast, only *GmMYB78*-GFP and nuclear marker *H2B*-mCherry fusion protein were observed in the nucleus. These results proved that GmMYB78 is a nucleus-localized transcription factor.

### 2.2. GmMYB78 Increases Susceptibility to P. sojae in Transgenic Soybean Hairy Roots 

To analyze the function of GmMYB78, the plant overexpressing vector *GmMYB78*-OE (35S:*GmMYB78*-4myc overexpression) and RNA interference vector *GmMYB78*-RNAi were constructed and transformed into susceptible soybean cultivar ‘Dongnong 50’ to produce transgenic soybean hairy roots by the high-efficiency *A. rhizogenes*-mediated transformation (Figure 2A). We characterized the *GmMYB78*-OE transgenic hairy roots using RT-qPCR analysis (Figure 2B) and western blotting (Appendix A), and the *GmMYB78*-RNAi transgenic hairy roots using qRT-PCR analysis and Quickstix Kits for Liberty Link (bar) strips (Figure 2B and Appendix A). After 72 hpi, the *GmMYB78*-OE transgenic lines displayed more serious disease symptoms than roots infected with the empty vector (EV), with the inoculation site and its surroundings being dark brown and having obvious soft rot (Figure 2A). In contrast, the *GmMYB78*-RNAi transgenic lines had no obvious symptoms compared with the EV (Figure 2A). The total area of lesions and the relative biomass content of *P. sojae* were significantly higher in the roots of the *GmMYB78*-OE lines compared with those infected with the EV (Figure 2C–E). By contrast, the area of lesions and relative biomass of *P. sojae* were significantly lower in the roots of the *GmMYB78*-RNAi lines compared with the EV control (Figure 2C–E). Taken together, these results indicated that *GmMYB78* is a negative regulator of the response to *P. sojae* infection.

### 2.3. GmMYB78 Inhibits Pathogenesis-Related (PR) Gene Expression in Response to P. sojae Infection 

The *PR* gene is an important part of the plant defense response to invasive pathogens, and MYB transcription factors can participate in the plant stress response by regulating the expression of *PR* genes [19,38]. To investigate the potential defense mechanism of the *GmMYB78*-regulated response to *P. sojae*, we examined the expression of some candidate *PR* genes *GmPR1* (GenBank accession no. AF136636), *GmPR2* (GenBank accession no. M37753), *GmPR3* (GenBank accession no. AF202731), and *GmPR10* (GenBank accession no. FJ960440). The results showed that the expression levels of *GmPR2*, *GmPR3*, and *GmPR10* were significantly lower than those of EV in *GmMYB78*-OE hairy roots, while the expression level of *GmPR1* was not significantly different from that of EV. In *GmMYB78*-RNAi hairy roots, the expression levels of *GmPR2*, *GmPR3*, and *GmPR10* showed the opposite results, and the expression level of *GmPR1* was not significantly different from that of EV (Figure 3A–D). The above results showed that *GmMYB78* responds to *P. sojae* by negatively regulating the expression of *GmPR2*, *GmPR3*, and *GmPR10*.

### 2.4. GmMYB78 Is a Negative Regulator of JA-Dependent Signaling during the Response to P. sojae

In order to further explore the susceptible mechanism of *GmMYB78* to *P. sojae* and explore its potential downstream target genes, in this study, the transcriptome (RNA-seq) and metabolome analysis were performed on *GmMYB78*-OE and EV transgenic soybean hairy roots. There were 617 differentially expressed genes (false-discovery rate set at <0.01, fold-change set at ≥2) identified between *GmMYB78*-OE and EV transgenic hairy roots, including 199 upregulated differentially expressed genes and 418 downregulated differentially expressed genes (Figure 4A,B). The GO functional analysis and KEGG enrichment indicated that these differentially expressed genes are involved in multiple life processes such as the plant hormone transduction pathway and stress response (Figure 4C,D). A metabolomics analysis showed that there were 96 differential metabolites (VIP ≥ 1, fold-change set at ≥2) identified between *GmMYB78*-OE and EV transgenic hairy roots, of which 17 were upregulated, and 79 were downregulated, including the jasmonic acid transduction pathway (Appendix A).

The combined analysis of the transcriptome and metabolome of *GmMYB78*-OE transgenic hairy roots showed that jasmonic acid (JA) was downregulated in the JA signal transduction pathway, and the expression of the key gene *GmAOS1* (*Allene oxide synthase 1*) (GenBank accession no. NP_001236432.1) in the JA synthesis pathway was also downregulated (Appendix A). MYB transcription factors play an important role in plant defense responses by regulating JA signaling [40,42,43]. In order to analyze the regulatory role of *GmMYB78* in the JA signaling pathway, we measured the JA content in *GmMYB78*-OE, EV, and *GmMYB78*-RNAi transgenic soybean hairy roots. The results showed that the JA content in *GmMYB78*-OE transgenic hairy roots was significantly lower than that in EV, while the JA content in *GmMYB78*-RNAi was significantly higher than that in EV hairy roots (Figure 5A). In addition, we also analyzed the transcriptional levels of the JA synthesis gene *GmAOS1*, and two negative regulators of the JA signaling pathway *GmJAZ1* (Phytozome. Glyma.07g041400) and *GmJAZ2* (Phytozome. Glyma16g010000) [44] in *GmMYB78* transgenic hairy roots by qRT-PCR. The results showed that the expression of *GmAOS1* was significantly lower in *GmMYB78*-OE transgenic hairy roots than that in EV, and significantly higher in *GmMYB78*-RNAi than that in EV roots (Figure 5B), while the expression of *GmJAZ1* and *GmJAZ2* was significantly higher in *GmMYB78*-OE transgenic hairy roots than that in EV, and significantly lower in *GmMYB78*-RNAi transgenic hairy roots than that in EV (Figure 5C,D). In summary, GmMYB78 can inhibit the JA signaling pathway by negatively regulating the JA synthesis gene *GmAOS1* expression and reducing the biosynthesis of JA, thereby negatively regulating soybean resistance to *P. sojae*.

Previous studies have shown that JA and salicylic acid (SA) signaling pathways sometimes have antagonistic effects in plants [45,46]. In this study, GmMYB78 inhibits the JA signaling pathway (Figure 5). In order to further clarify whether GmMYB78 is involved in the SA signaling pathway, we analyzed the transcriptional level of the SA synthesis gene *GmICS* (GenBank no. XM_003522145) and the salicylic acid signaling pathway gene *GmPAL* (GenBank no. NM_001250027) in *GmMYB78* transgenic hairy roots. The results showed that the expression levels of *GmICS* and *GmPAL* displayed no significant differences between *GmMYB78*-OE/*GmMYB78*-RNAi transgenic hairy roots and EV roots (Appendix A), suggesting that GmMYB78 is not involved in the SA signaling pathway.

### 2.5. GmMYB78 Regulates the Transcription of GmbZIP25

To test the activation of the transcription function of *GmMYB78*, we performed a transient expression assay in yeast cells using a GAL4-responsive reporter system. Transformed yeast cells harbouring DBD-P53 + T-antigen (pGBKT7-53 + pGADT7-T, positive control), BD-*GmMYB78* (pGBKT7-*GmMYB78*), and DBD-GmWRKY31 (pGBKT7-GmWRKY31) [47], which exhibited a transcriptional activation ability in our previous studies, grew well in synthetic dropout medium without tryptophan, histidine, and adenine [SD (-Trp/-His/-Ade)] and showed α-galactosidase (α-gal) activity, whereas yeast cells that were empty (pGBKT7, negative control) exhibited no α-gal activity (Figure 6A). These data confirm that GmMYB78 has transcriptional activation activity.

In order to further analyze the downstream genes activated by GmMYB78, we selected eight stress-related upregulated differentially expressed genes based on the RNA-seq for validation by RT-qPCR in *GmMYB78*-OE and *GmMYB78*-RNAi transgenic hairy roots. Of the eight genes tested, the expression levels of *GmPRR13*, *GmbZIP25*, *GmMYB177*, *GmLOX9*, *GmCHS13*, and *GmARM* were upregulated (Figure 6B). On the contrary, the expression levels of *GmPRR13*, *GmbZIP25*, *GmMYB177*, *GmLOX9*, *GmCHS13*, and *GmARM* were downregulated in *GmMYB78*-RNAi transgenic hairy roots (Figure 6C). In addition, *GmbZIP25* transcript levels increased the most in the *GmMYB78*-OE hairy roots compared to EV (Figure 6B) and showed the strongest decrease in *GmMYB78*-RNAi (Figure 6C). The above results suggest that *GmbZIP25* may be a downstream target of GmMYB78 in defense responses.

MYB transcription factors can specifically bind to MBS (MYB-binding sites) cis-acting elements (T/C) AAC (G/T) G (A/C/T) (A/C/T), (C/T) NGTT (A/G), ACC (A/T) A (A/C) (T/C), and ACC (A/T) (A/C/T) (A/C/T) (A/C/T) in the target gene promoter, and then participate in plant growth and metabolic pathways and respond to various biotic and abiotic stress defense responses [48,49,50,51]. We analyzed the cis-acting elements in the promoter of *GmbZIP25* and found that *pGmbZIP25* contained two MYB transcription factor-specific binding motifs, i.e., CAACGGAT and ACCAAC. Therefore, we further analyzed the binding ability of GmMYB78 to the *GmbZIP25* promoter. Firstly, the 2000 bp upstream promoter of *GmbZIP25* was cloned, and a 387 bp–797 bp interval, which contained the MYB transcription factor-specific binding motifs CAACGGAT and ACCAAC, was isolated. The recombinant vector *pGmbZIP25*-pBait-AbAi was constructed according to the 387 bp–797 bp interval sequence. The recombinant vector was transformed into yeast Y1H Gold and the concentration of AbA inhibiting its yeast growth strain was screened. The results showed that, when the AbA concentration was 400 ng/mL, the growth of the bait strain Y1H [pBait-AbAi-*pGmbZIP25*] was completely inhibited (Appendix A), so the yeast one-hybrid test was carried out at this concentration. 

Next, the recombined vector *GmMYB78*-AD was transformed into the Y1H Gold [pBait-AbAi-*pGmbZIP25*] yeast strain, with pGADT7-53 or pGADT7 transformed into the yeast cells Y1H Gold [p53-AbAi] as positive and negative controls, respectively. They were then sequentially spread on SD/-Leu and SD/-Leu + AbA_400_ agar plates and cultured upside down at 30 °C for 3–7 days. As shown in Figure 6D, all yeast cells grew normally on the SD/-Leu agar plates. The positive control and the *GmMYB78*-AD-fused Y1H Gold [pBait-AbAi-*pGmbZIP25*] yeast cells grew normally, while the negative control did not grow on the SD/-Leu + AbA_400_ agar plates. The results indicated that GmMYB78 can directly bind to the promoter of *GmbZIP25*.

Finally, the mechanism of GmMYB78 regulating *GmbZIP25* expression was explored by the luciferase reporter system. We cloned the full-length (2000 bp) promoter of *GmbZIP25* and constructed the reporter vectors *p35S:Rluc-pGmbZIP25:LUC*. The effector construct harboring GmMYB78 was expressed under the control of the 35S promoter (*p35S: GmMYB78-Myc*), and an EV control (Figure 6E). The following combinations were each co-transformed into *N. benthamiana* leaves: *p35S:Rluc-pGmbZIP25:LUC* + EV and *p35S:Rluc-pGmbZIP25:LUC* + *p35S: GmMYB78-Myc*. After 3 d, 1 mM D-Luciferin was sprayed onto the transformed leaves and imaged using a chemiluminescence system. Compared with the leaves transfected with *p35S:Rluc-pGmbZIP25:LUC* + EV, *p35S:Rluc-pGmbZIP25:LUC* + *p35S: GmMYB78-Myc* displayed a significantly enhanced chemiluminescence signal (Figure 6F). The results of the LUC/Rluc relative activity assay further indicated that GmMYB78 could directly enhance *GmbZIP25* expression (Figure 6G). The above results indicated that GmMYB78 can directly bind to the promoter of *GmbZIP25* to enhance its expression.

### 2.6. GmbHLH122 Regulates the Transcription of GmMYB78

In our previous study, *GmMYB78* was downregulated in *GmAP2*-OE hairy roots according to the transcriptome sequencing analysis [41]. To determine whether *GmMYB78* is a downstream target gene directly regulated by GmAP2, the binding of GmAP2 to the *GmMYB78* promoter was analyzed by the yeast one-hybrid. Firstly, the AbA concentration that inhibited Y1H [pBait-AbAi-*pGmMYB78*] was determined to be 500 ng/mL (Appendix A). Further yeast one-hybrid experiments showed that the positive control yeast cells grew normally, while Y1H Gold [pBait-AbAi-*pGmMYB78*] fused with *GmAP2*-AD and the negative control did not grow on SD/-Leu + AbA_500_ agar plates, indicating that *GmMYB78* is not a direct downstream target gene of GmAP2 (Appendix A). In addition, the luciferase reporter system analysis results indicated that, compared with the co-transfection of *N. benthamiana* leaves with the reporter vector *p35S:Rluc-pGmMYB78:LUC* and EV, there was no significant difference in the chemiluminescence signal of *N. benthamiana* leaves co-transfected with the reporter vector *p35S:Rluc-pGmMYB78:LUC* and the effector vector *p35S:GmAP2-Myc* (Appendix A). Furthermore, the relative activity of LUC/REN further proves that *GmMYB78* is not a direct downstream target gene regulated by GmAP2 (Appendix A). 

In order to further determine the upstream regulatory factors of *GmMYB78*, we used a yeast one-hybrid (Y1H) screen of a cDNA library from the soybean disease-resistant cultivar ‘Suinong 10’ infected with *P. sojae* based on the promoter (113 bp–670 bp) of *GmMYB78*. The ESTs from four candidate genes encoding proteins that might be regulatory factors of *GmMYB78* are listed in Appendix A, of which a bHLH transcription factor GmbHLH122 (XM-003543501.5) was appraised. In this study, we further analyzed the transcriptional regulation of *GmMYB78* by GmbHLH122 by the yeast one-hybrid. The recombined vector *GmbHLH122*-AD was transformed into the Y1H Gold [pBait-AbAi-*pGmMYB78*] yeast strain, with pGADT7-53 or pGADT7 transformed into the yeast cells Y1H Gold [p53-AbAi] as positive and negative controls, respectively. They were then sequentially spread on SD/-Leu and SD/-Leu + AbA_500_ agar plates and cultured upside down at 30 °C for 3–7 days. As shown in Figure 7A, all yeast cells grew normally on the SD/-Leu agar plates. The positive control and the *GmbHLH122*-AD-fused Y1H Gold [pBait-AbAi-*pGmMYB78*] yeast cells grew normally, while the negative control did not grow on the SD/-Leu + AbA_500_ agar plates. The results indicated that GmbHLH122 can directly bind to the promoter of *GmMYB78*.

Next, the mechanism of GmbHLH122 regulating *GmMYB78* expression was explored by the luciferase reporter system. We constructed the reporter vectors *p35S:Rluc-pGmMYB78:LUC*, the effector construct harboring GmbHLH122 expressed under the control of the 35S promoter (*p35S:GmbHLH122-Myc*), and an EV control (Figure 7B). The following combinations were each co-transformed into *N. benthamiana* leaves: *p35S:Rluc-pGmMYB78:LUC* + EV and *p35S:Rluc-pGmMYB78:LUC* + *p35S:GmbHLH122-Myc*. After 3 d, 1 mM D-Luciferin was sprayed onto the transformed leaves and imaged using a chemiluminescence system. Compared with the leaves transfected with *p35S:Rluc-pGmMYB78:LUC* + EV, *p35S:Rluc-pGmMYB78:LUC* + *p35S:GmbHLH122-Myc* displayed a significantly suppressed chemiluminescence signal (Figure 7C). The results of the LUC/Rluc relative activity assay further indicated that GmbHLH122 could directly inhibit *GmMYB78* expression (Figure 7D). The above results indicated that GmbHLH122 can directly bind to the promoter of *GmMYB78* to inhibit its expression. 

### 2.7. Expression Analysis of GmbLHL122, GmMYB78, and GmbZIP25 in Response to P. sojae

In order to further elucidate the transcriptional regulatory mechanism of the GmbLHL122–GmMYB78-*GmbZIP25* module in response to *P. sojae*, we measured the expression levels of *GmbHLH122*, *GmMYB78*, and *GmbZIP25* in the susceptible soybean cultivar ‘Dongnong 50’ and the resistant soybean cultivar ‘Suinong 10’ upon *P. sojae* infection. In ‘Dongnong 50’, *GmbHLH122* expression was not obviously activated by *P. sojae* infection, while *GmMYB78* expression was quickly upregulated, reaching a peak at 9 h post-inoculation (hpi); *GmbZIP25* was also upregulated and peaked at 12 hpi (Figure 8A). In ‘Suinong 10’, infection by *P. sojae* quickly induced the upregulation of *GmbHLH122* expression, which reached a peak at 12 hpi, while the expression of *GmMYB78* and *GmbZIP25* were not obviously activated by *P. sojae* infection (Figure 8B). These results suggest that *GmbLHL122* is highly expressed in resistant soybean cultivars, while *GmMYB78* and *GmbZIP25* are more highly expressed in susceptible soybean cultivars following *P. sojae* infection.

## 3. Discussion 

### 3.1. GmMYB78 Negatively Regulates P. sojae Infection by Inhibiting the Expression of Pathogenesis-Related Genes

Transcription factors MYB play important roles in plants responding to fungal, bacterial, and oomycete infections [28,31,32,52,53,54]. For example, the overexpression of *AtMYB44* weakens the Arabidopsis defense response against *Pseudomonas syringae* pv. tomato DC3000 [31]. VaMYB306 can form a transcription complex with VaERF16 to enhance the transcription of the defense-related gene *VaPDF1.2*, thereby positively regulating resistance to *Botrytis cinerea* [55]. Our previous studies demonstrated that *GmBTB/POZ* can enhance soybean resistance to *P. sojae* [56], and an AP2/ERF transcription factor GmAP2 was identified by the yeast two-hybrid, which negatively regulates soybean resistance to *P. sojae* [41]. In this study, RNA-seq analysis of GmAP2 identified a downregulated MYB family transcription factor, GmMYB78. In order to explore whether GmMYB78 is a potential downstream target gene of GmAP2, we used the yeast one-hybrid and dual luciferase reporter system to verify it. The results showed that GmAP2 does not directly regulate the expression of *GmMYB78* (Appendix A), suggesting GmMYB78 may indirectly participate in the defense response of soybean to *P. sojae* through GmAP2. To further explore whether *GmMYB78* is involved in the response to *P. sojae* infection in soybean, transgenic soybean hairy roots of *GmMYB78* were analyzed for disease resistance, indicating that *GmMYB78* is a negative regulator in response to *P. sojae* infection (Figure 2). Multiple homologous genes of GmMYB78 were also identified in other species [57,58,59,60], among which JAMYB in *Medicago truncatula* was identified as a candidate gene related to *Erysiphe pisi* resistance [59].

*PR* genes are some of the most important genes in the plant defense response to pathogens [61,62,63,64]. Transcription factors can respond to pathogens by regulating the expression of *PR* genes [65,66]. Studies have found that soybean *GmERF113* can enhance transgenic soybean resistance to *P. sojae* by positively regulating the expression of *GmPR1* and *GmPR10-1* [67]. GmNPR1 enhances soybean resistance to *P. sojae* by activating the expression of *GmPR1a*, *GmPR2*, *GmPR3*, and *GmPR10* [47]. To investigate whether GmMYB78 can also participate in the response of soybeans to *P. sojae* by affecting the expression of *PR* genes, we analyzed the expression levels of *PR* genes in *GmMYB78* transgenic hairy roots. The results indicate that GmMYB78 can downregulate the expression of *GmPR2*, *GmPR3*, and *GmPR10*, rather than *GmPR1*, reducing soybean resistance to *P. sojae*. *PR1* usually act as effector genes for systemic acquired resistance (SAR), a process mediated by salicylic acid (SA); high expression levels of these genes indicate the activation of SA signaling [68,69]. Since GmMYB78 is not involved in the SA signaling pathway, we speculate that one of the reasons why GmMYB78 cannot regulate *GmPR1* expression is its inability to respond to SA signaling. These findings suggested that GmMYB78 can reduce the resistance to *P. sojae* through the negative regulation of pathogenesis-related gene expression.

### 3.2. GmMYB78 Is Involved in Soybean Defense Response to P. sojae through JA Signaling Pathway

The JA signaling pathway plays an important role in plant immunity and plants can respond to biotic stresses through the JA signaling pathway [40,44,70]. For example, Avh94 can interact with the JA signaling repressor JAZ1/2, stabilizing JAZ1/2 to inhibit JA signal transduction, thereby negatively regulating soybean resistance to *P. sojae* [44]. The overexpression of *OsMPK15* can lead to the significant downregulation of SA and JA-related genes, thereby negatively regulating rice resistance to *Magnaporthe oryzae* [71]. Cotton GhJAZ2 can interact with GhbHLH171 and inhibit its transcriptional activity, thereby inhibiting JA synthesis, reducing plant tolerance to *Verticillium dahliae* [72]. Rice OsMYC2 activates its expression by binding to the G-box motif in the *OsJAZ10* promoter, enhancing sensitivity to JA and positively regulating rice resistance to *Xanthomonas oryzae* pv. *oryzae (Xoo)* [73].

It has been reported that MYB transcription factors can participate in plant responses to biotic stress through the JA signaling pathway [40,74,75]. For example, silencing the MYB transcription factor *GhODO1* impairs JA-mediated defense signals, inhibits the expression of genes in the JA synthesis pathway, thereby reducing JA accumulation, and negatively regulates the resistance of cotton (*Gossypium hirsutum*) to *Verticillium dahlia* [43]. RcMYB84 interacts with the key repressor of JA signaling, JAZ1, and their complex binds to the *RcMYB123* promoter through the CAACTG motif to inhibit its transcription, thereby reducing rose resistance to *Botrytis cinerea* [76]. In this study, we found that JA was downregulated in the metabolome of *GmMYB78* and the key gene for JA synthesis, *GmAOS*1, showed a downregulated expression in the RNA-seq assay of GmMYB78. Further research found that the JA content was decreased and the expression of *GmAOS1* was downregulated in *GmMYB78*-OE transgenic hairy roots. The JA content and JA pathway-related gene expression in *GmMYB78* transgenic hairy roots confirmed that *GmMYB78* can reduce soybean resistance to *P. sojae* by negatively regulating JA synthesis and signal transduction. Research has shown that the JA and salicylic acid (SA) signaling pathways can sometimes have antagonistic effects within plant bodies [45,46]. Therefore, the expression levels of genes related to the SA signaling pathway were analyzed in *GmMYB78* transgenic soybean hairy roots. The results demonstrate that GmMYB78 does not affect the SA synthesis or gene expression involved in its signaling transduction pathway (Appendix A). Furthermore, SA signaling pathway genes and related metabolites were not found in the transcriptome and metabolome analysis of GmMYB78, leading us to speculate that GmMYB78 is not involved in the SA signaling transduction pathway. These results indicating that *GmMYB78* can reduce resistance to *P. sojae*, potentially functioning via effects on the expression of JA-related genes and reduced accumulation of JA.

### 3.3. GmbHLH122-GmMYB78-GmbZIP25 Regulatory Module Is Involved in the Response to P. sojae in Soybean

Research has shown that MYB transcription factors can be regulated as downstream target genes of defense-related genes, thereby responding to environmental stress [77,78]. The wheat stripe rust resistance transcription factor TaWRKY10 may bind to the W-Box elements on the promoters of MYB transcription factors *TaLHYa*, *TaLHYb*, and *TaLHYd* to regulate the expression of *TaLHY* [79]. In order to identify the upstream protein that directly regulates *GmMYB78*, this study used the *GmMYB78* promoter region as bait protein to screen for candidate upstream regulatory genes from a yeast one-hybrid cDNA library induced by *P. sojae*, and preliminary evidence through the yeast one-hybrid and dual-luciferase reporter systems demonstrated that GmbHLH122 can directly bind to the *GmMYB78* promoter and suppress its expression (Figure 7). bHLH transcription factors are widely involved in the response to biotic stress in plants [8,41]. For example, the bHLH transcription factor GmPIB1 can significantly reduce the accumulation of ROS (reactive oxygen species) and directly bind to the cis-acting elements in the promoter of *GmSPOD1* to inhibit its expression, positively regulating soybean resistance to *P. sojae* [8]. The overexpression of the bHLH transcription factor *GhbHLH171* can activate JA synthesis signaling pathways, thereby enhancing the tolerance to *Verticillium dahliae* in cotton [72]. In this study, *GmMYB78* is a negative regulator in soybean response to *P. sojae* infection, and GmbHLH122, as a direct regulator of *GmMYB78*, was induced by *P. sojae* infection in resistant cultivar ‘Suinong 10’ (Figure 8B) and may be involved in the resistance to *P. sojae* infection in soybean, which will be the focus of future research.

MYB transcription factors activate or inhibit the expression of downstream defense-related genes by specifically binding to the promoters, thus participating in the regulation of plant defense responses [48,49,50]. For example, TuMYB46L can bind to the promoter region of the ethylene synthesis-related gene *TuACO3*, inhibiting ethylene biosynthesis and negatively regulating wheat resistance to powdery mildew [30]. BjMYB1 can bind to chitinase *BjCHI1* promoter, activating *BjCHI1* expression to enhance transgenic Arabidopsis resistance to *Botrytis cinerea* [80]. CsMYB96 directly binds to the *CsCBP60g* (calmodulin binding protein 60 g) promoter, activating SA signal transduction to enhance resistance of citrus and Arabidopsis to fungal pathogens [81]. This study demonstrates that GmMYB78 is located in the nucleus (Figure 1B) and can function as a transcriptional activation factor to activate the transcription of reporter genes (Figure 6A). In addition, the transcriptome sequencing of *GmMYB78* transgenic soybean hairy roots revealed that GmMYB78 can activate the expression of multiple stress-responsive genes including *GmbZIP25* (Figure 6B,C), with *GmbZIP25* showing the most significant change in expression in *GmMYB78* transgenic hairy roots, thus further validating *GmbZIP25* as a downstream target gene directly regulated by GmMYB78. An analysis of the *GmbZIP25* promoter revealed the presence of two MYB binding motifs in its promoter region. Further yeast one-hybrid and dual-luciferase reporter system assays confirmed that GmMYB78 can directly bind to the *GmbZIP25* promoter and activate its expression (Figure 6D–F). It has been reported that bZIP transcription factors, as important defense signaling genes, are widely involved in plant defense responses to biotic stresses [82]. CsAtf1, bZIP transcription factor, can enhance the virulence of *Colletotrichum siamense* by regulating the expression of *CsPbs2* and *CsHog1* in the MAPK pathway, thereby reducing rubber tree resistance to anthracnose [83]. Knocking out *TabZIP74* increased the susceptibility of wheat seedlings to stripe rust [84]. Silencing *RcbZIP17* can reduce the resistance of roses to *Botrytis* [85]. In this study, *GmMYB78* is a negative regulator in soybean response to *P. sojae* infection, and *GmbZIP25*, as a target gene directly activated by GmMYB78 (Figure 6D–G), was induced by *P. sojae* infection in susceptible cultivar ‘Dongnong 50’ (Figure 8A) and may also be involved in the defense response of soybean to *P. sojae*. We will further verify the binding of GmMYB78 to the promoter region of *GmbZIP25* and analyze the function of *GmbZIP25* in the future works of research. This will reveal the role of the GmbHLH122-GmMYB78-*GmbZIP25* regulatory pathway in the soybean response to *P. sojae* infection, providing a theoretical basis for unraveling the molecular mechanisms of GmMYB78 in response to *P. sojae*.

Based on these results, we propose a model that explains the mechanisms of *GmMYB78* response to *P. sojae* infection (Figure 9). According to our model, in the susceptible cultivar ‘Dongnong 50’, *GmbHLH122* is not activated as a result of *P. sojae* infection, while *GmMYB78* is upregulated and accumulated, and represses downstream *GmbZIP25* expression, downregulates the expression of *GmPR* genes and JA-synthesis-pathway-related genes, and reduces JA content, thereby inhibiting the plant defense responses (left); in the resistant cultivar ‘Suinong 10’, *GmbHLH122* transcription is induced by *P. sojae* infection. The high levels of GmbHLH122 inhibits the transcription of *GmMYB78*, which reduce its promotive effect on *GmbZIP25* expression, upregulates *GmPR* genes and JA-synthesis-pathway-related genes, and promotes the accumulation of JA content, thereby enhancing the defense response to *P. sojae* (right). Thus, our study provides novel insights into the mechanisms by which the GmbHLH122-GmMYB78-*GmbZIP25* pathway modulates the defense response of soybean during infection by *P. sojae*. 

## 4. Materials and Methods

### 4.1. Plant Materials and Growth Conditions

This study used ‘Dongnong 50’ (highly susceptible to *Phytophthora sojae* race 1, the predominant race in Heilongjiang, China), and ‘Suinong 10’, carrying resistance against *P. sojae* [86]. ‘Dongnong 50’ and ‘Suinong 10’ seeds were sown in soil and grown in a growth chamber at 28 °C and 70% relative humidity under a 16 h light/8 h dark photoperiod. These plants were used for gene isolation and analysis of relative transcript levels [87,88] when the first true leaf was about to unfold (V1) [89]. *Phytophthora sojae* No.1 physiological race was preserved in the Key Laboratory of Soybean Biology, Ministry of Education for identification of pathogen inoculation. *Arabidopsis thaliana* seeds were grown in a growth chamber at 23 °C under an 8 h light/16 h dark photoperiod. Arabidopsis plant used for subcellular localization assay before bolting. *Nicotiana benthamiana* seeds were grown in a growth chamber at 28 °C under a 16 h light/8 h dark photoperiod. *N. benthamiana* plants at 6 weeks old were used for dual-luciferase assays.

### 4.2. RT-qPCR

Total RNA was isolated from leaves or hairy roots of soybean using Trizol reagent (Invitrogen, Shanghai, China), and RT-qPCR analysis was performed on a LightCycler96 instrument (Roche, Switzerland) with a real-time PCR kit (TOYOBO, Osaka, Japan). The gene expression levels were calculated by the 2^−ΔΔCt^ method with *GmEF1β* (GenBank accession no. NM_001248778) and *GmTUB4* (GenBank accession no. EV263740) as the internal control. The reaction conditions consisted of an initial 5 min pre-incubation at 94 °C, and 40 cycles at 94 °C for 30 s, 59 °C for 30 s, and 72 °C for 40 s, followed by a melting-curve analysis from 55 °C to 100 °C with a final cooling for 10 min at 72 °C. The relative transcript abundance of each target gene was calculated using the 2^−∆∆CT^ method. The primers used for expression analysis are shown in Appendix A.

### 4.3. Gene Cloning, Sequence Analyses, and Plasmid Construction

The full-length *GmMYB78*, *GmbHLH122*, and *GmbZIP25* genes were amplified by PCR using the gene-specific primers (Appendix A). Sequence alignments were performed using DNAMAN (http://www.lynnon.com/, accessed on 3 December 2021). Molecular Evolutionary Genetics Analysis (MEGA) software 5.1 were used to analyze phylogenetic of GmMYB78 and other MYBs. The ORF of *GmMYB78* was cloned into the vector pCAMBIA3301 with the *bar* gene and 4 × Myc tag as the selectable marker under the control of the cauliflower mosaic virus 35S (CaMV35S) promoter to overexpress the *GmMYB78* gene. *GmMYB78*-RNAi constructs were created based on pFGC5941 vector following the methods described by Kerschen et al. [90].

### 4.4. Subcellular Localization Assays of the GmMYB78 Protein

The full-length coding sequence of *GmMYB78* was cloned into pCAMBIA1302 (GFP) vector. The coding sequence was placed under the control of the CaMV 35S promoter and cloned in-frame and upstream of the sequence encoding green fluorescent protein (GFP). The resulting *35S:GmMYB78-GFP* expression plasmid and *35S:GFP* control were individually co-transfected with nuclear Marker H2B-mCherry into Arabidopsis protoplasts using PEG-mediated transfection, as described by Yoo et al. [91]. The transformed protoplasts were observed on a confocal spectral microscope imaging system (Leica TCS SP8, Wetzlar, Germany).

### 4.5. Agrobacterium Rhizogenes-Mediated Transformation of Soybean Hairy Roots

The recombinant constructs of *GmMYB78*-OE and *GmMYB78*-RNAi were introduced into *Agrobacterium rhizogenes* K599 strain and transformed into susceptible soybean cultivar ‘Dongnong 50’ to induce transgenic soybean hairy roots according to a modified method described by Graham et al. [92] and Kereszt et al. [93]. Briefly, we removed the growth point of soybean cotyledon and cut the wound on the surface of cotyledons. *Agrobacterium rhizogenes* bacteria carrying the target gene plasmid were cultured to OD_600_ = 0.6 and the harvested cells were resuspended. *A. rhizogenes* suspension was applied to the cut surface of hypocotyls. The treated soybean seedlings were put on root-inducing medium in dishes and placed in an incubator at 25 °C for 3 weeks. The empty vectors were used as controls. The transgenic hairy taproots will be used for *P. sojae* inoculation. The lateral roots grown on the taproots were tested to confirm the positive taproots. The transgenic hairy taproots with overexpression-target gene were tested via immunoblots, and RNAi transgenic hairy taproots were verified by QuickStix Kit for LibertyLink (bar) strip detection. The expression level of the overexpression and interference genes were also detected by RT-qPCR. The healthy taproots with similar size were selected for *P. sojae* infection. 

### 4.6. Assessment of Soybean Disease Responses

The pathogen resistances of the *GmMYB78*-OE, *GmMYB78*-RNAi, and empty vector (EV) transgenic soybean hairy root lines were assessed using artificial inoculation assays with *P. sojae*, as described by Ward et al. [87] and Dou et al. [94], with minor modifications. When the hairy roots generated at the infection site grew approximately for 3 weeks, which grew to around 7 cm and were detected as positive ones, the tap roots were incubated with *P. sojae* zoospores (approximately 1 × 10^5^ spores mL^−1^) in a mist chamber at 25 °C with 100% relative humidity for 72 h. EV soybean hairy roots were used as controls. Disease symptoms were imaged using a Nikon B7000 camera at 72 h after inoculation, and the total area of lesions was determined using ImageJ software, https://imagej.net/ij/index.html, accessed on 8 January 2024). 

### 4.7. Determination of JA Levels

Jasmonic acid (JA) was extracted from *GmMYB78* or EV transgenic soybean hairy roots and quantified using HPLC-mass spectrometry, as described by Zhu et al. [95].

### 4.8. RNA-Seq and Metabolome Analysis

The hairy roots of *GmMYB78*-OE and the EV control grown under non-stress conditions were used for RNA-seq or metabolites analysis. For RNA-seq assay, the cDNA libraries were sequenced on the Illumina sequencing platform by Metware Biotechnology Co., Ltd. (Wuhan, China). We used HISATv2.1.0 to construct the index, and compared clean reads to the reference genome (https://phytozome.jgi.doe.gov/, accessed on 12 January 2022). DESeq2 v1.22.1 was used to analyze the differential expression between *GmMYB78*-OE and EV transgenic soybean hairy roots. During differential expression gene detection, fold change ≥1, FDR < 0.05 was used as a screening standard. The enrichment analysis is performed based on the hypergeometric test. For KEGG, the hypergeometric distribution test is performed with the unit of pathway; for GO, it is performed based on the GO term. 

For metabolome assay, significantly regulated metabolites between groups were determined by VIP ≥ 1 and absolute log2FC (fold change) ≥ 1. Identified metabolites were annotated using KEGG compound database (http://www.kegg.jp/kegg/compound/, accessed on 4 February 2022); annotated metabolites were then mapped to KEGG pathway database (http://www.kegg.jp/kegg/pathway.html, accessed on 4 February 2022). Pathways with significantly regulated metabolites mapped to them were then fed into MSEA (metabolite sets enrichment analysis); their significance was determined by hypergeometric test’s *p*-values.

### 4.9. Yeast One-Hybrid (Y1H) Assays

Yeast one-hybrid assays were used to examine the binding of GmbHLH122 to the *GmMYB78* promoter and GmMYB78 to the *GmbZIP25* promoter. The coding sequence of *GmbHLH122* or *GmMYB78* was cloned into the pGADT7 plasmid, while the promoter of *GmMYB78* or *GmbZIP25* was inserted into the pBait-AbAi plasmid. The background leakiness of the pBait-AbAi-*pGmMYB78* or pBait-AbAi-*pGmbZIP25* recombinant plasmid was suppressed by adding AbA to the synthetic defined (SD) medium lacking -Ura (SD/-Ura) medium. The recombinant vector *GmbHLH122*-AD or *GmMYB78*-AD was transformed into Y1H [pBait-AbAi-*pGmMYB78*] or Y1H [pBait-AbAi-*pGmbZIP25*] yeast cells and spotted onto synthetic defined (SD) medium lacking -Leu (SD-Leu) or SD/-Leu/AbA to determine the interactions. The pGADT7-P53 and pGADT7 plasmids were used as positive and negative controls, respectively.

### 4.10. Transient Transcription Dual-Luciferase Assays

The promoter region of *GmMYB78* or *GmbZIP25* was amplified, cloned into the pGreenII 0800-LUC vector, and used as the reporter. The *35S:GmbHLH122-Myc* or *35S:GmMYB78-Myc* construct was used as effector. The effector and reporter constructs were introduced into Agrobacterium (*Agrobacterium tumefaciens* strain GV3101). Mixtures of agrobacteria resuspended in infiltration buffer in a 1:1 ratio were infiltrated into healthy leaves of 28-day-old *N. benthamiana* leaves [96]. The *N. benthamiana* were incubated under continuous dark light for 2 days and white light for 1 day after infiltration, sprayed with luciferin (1 mM luciferin and 0.01% [*v*/*v*] Triton X-100), and photographed using a chemiluminescence imaging system (Tanon 5200). Leaf samples were collected to measure firefly luciferase (LUC) and Renilla luciferase (Rluc) activities with a commercial dual-luciferase assay kit (Promega; PR-E1910). The *Rluc* in the pGreenII 0800-LUC vector was used as an internal control. LUC activity was normalized to Rluc activity. 

### 4.11. Statistical Analysis

All experiments were performed at least three times and significant differences between means were determined using Student’s *t*-test.

## Figures and Tables

**Figure 1 ijms-25-04247-f001:**
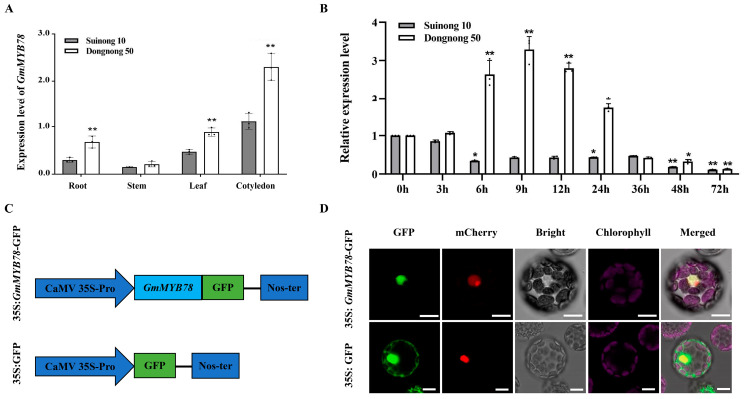
GmMYB78 is localized in the nucleus and induced by *P. sojae.* (**A**) Spatial expression patterns of *GmMYB78* in the resistant soybean cultivar ‘Suinong 10’ and the susceptible cultivar ‘Dongnong 50’ under normal conditions. (**B**) Relative expression of *GmMYB78* in the resistant soybean cultivar ‘Suinong 10’ and susceptible ‘Dongnong 50’ during *P. sojae* infection. The infected samples were collected at 0, 3, 6, 9, 12, 24, 36, 48, and 72 h after *P. sojae* infection (hpi). The reference gene *GmEF1β* (GenBank accession no. NM_001248778) and *GmTUB4* (GenBank accession no. EV263740) were used as internal control. The experiment was performed on three biological replicates, each with three technical replicates, and statistical significance was analyzed using Student’s *t*-test (* *p* < 0.05, ** *p* < 0.01). Bars indicate the standard error of the mean. (**C**) Schematic representations of *P35S:GmMYB78*-GFP or *P35S:*GFP vector. (**D**) *GmMYB78*-GFP or GFP was co-transfected into Arabidopsis protoplasts with nuclear Marker gene *H2B*-mCherry, respectively, and the fluorescence signals were observed under a confocal microscope. Bars indicate 10 µm.

**Figure 2 ijms-25-04247-f002:**
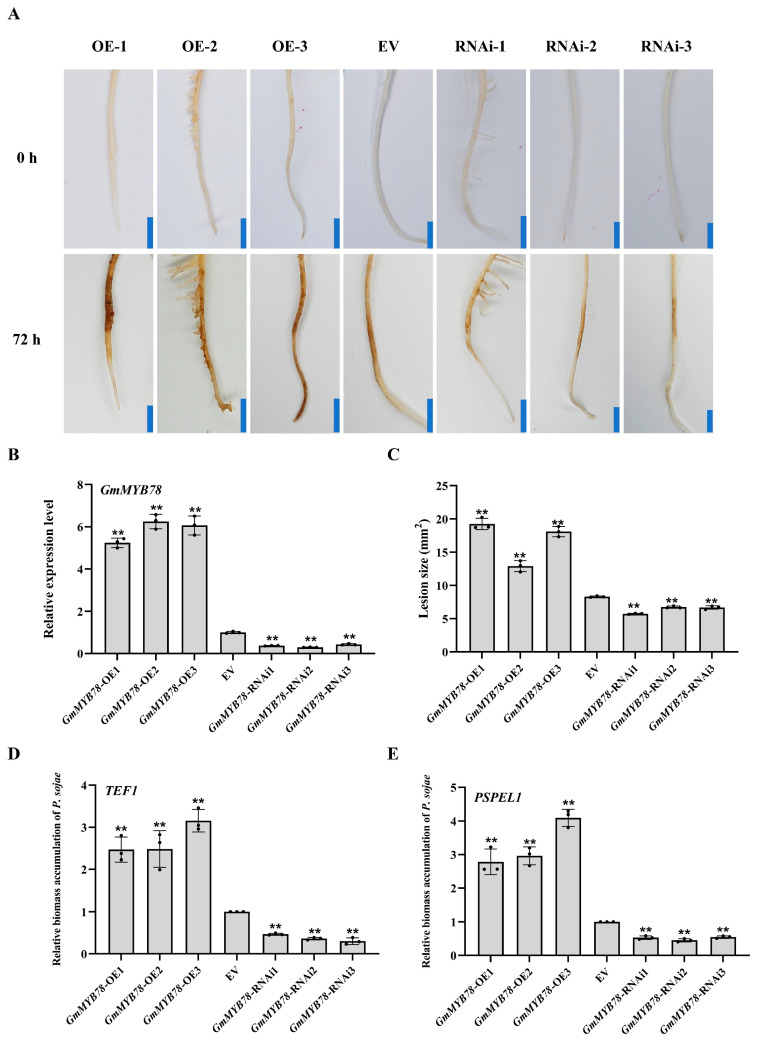
*GmMYB78* is a negative regulator in response to *P. sojae.* (**A**) Symptoms of *GmMYB78*-OE, *GmMYB78*-RNAi, and empty vector (EV) soybean hairy roots inoculated by *P. sojae* at 72 hpi. Bars = 0.5 cm. (**B**) Relative expression of *GmMYB78* in three *GmMYB78*-OE, *GmMYB78*-RNAi, and EV transgenic soybean hairy root as determined by qRT-PCR. *GmEF1β* and *GmTUB4* were used as the reference genes and expression is relative to that of the EV controls, the values of which were set as 1. (**C**) Lesion size in *GmMYB78*-OE, *GmMYB78*-RNAi, and EV transgenic soybean hairy roots at 72 hpi. (**D**,**E**) Relative biomass accumulation of *P. sojae* in the *GmMYB78*-OE, *GmMYB78*-RNAi, and EV roots based on transcript levels of (**D**) the fungal *TEF1* gene and (**E**) the *PSEL1* (*P. sojae* elicitin gene 1) gene. Fungal biomass is relative to that of the plant. The experiment was performed on three biological replicates, each with three technical replicates, and statistical significance was analyzed using Student’s *t*-test (** *p* < 0.01). Bars indicate the standard error of the mean.

**Figure 3 ijms-25-04247-f003:**
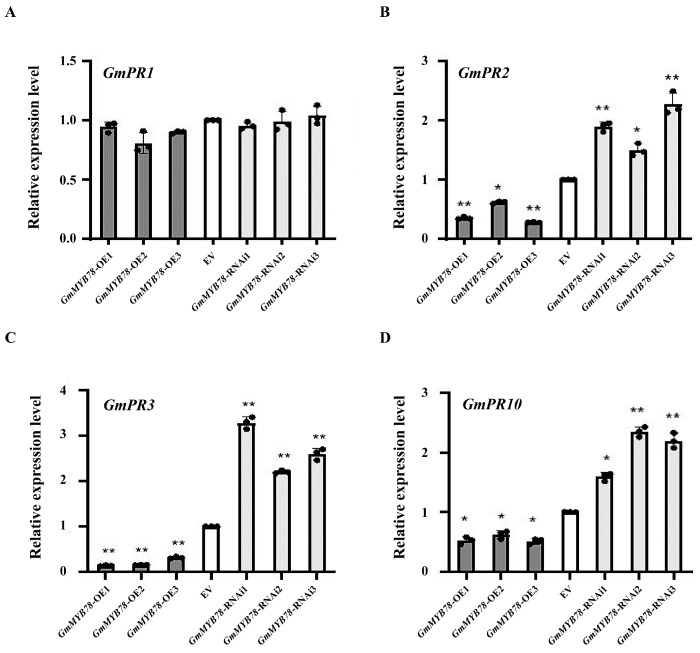
Relative expression levels of pathogenesis-related genes *GmPR1* (**A**), *GmPR2* (**B**), *GmPR3* (**C**), and *GmPR10* (**D**) in *GmMYB78* transgenic soybean hairy roots. The experiment was performed on three biological replicates, each with three technical replicates, and statistical significance was analyzed using Student’s *t*-test (* *p* < 0.05, ** *p* < 0.01). Bars indicate the standard error of the mean.

**Figure 4 ijms-25-04247-f004:**
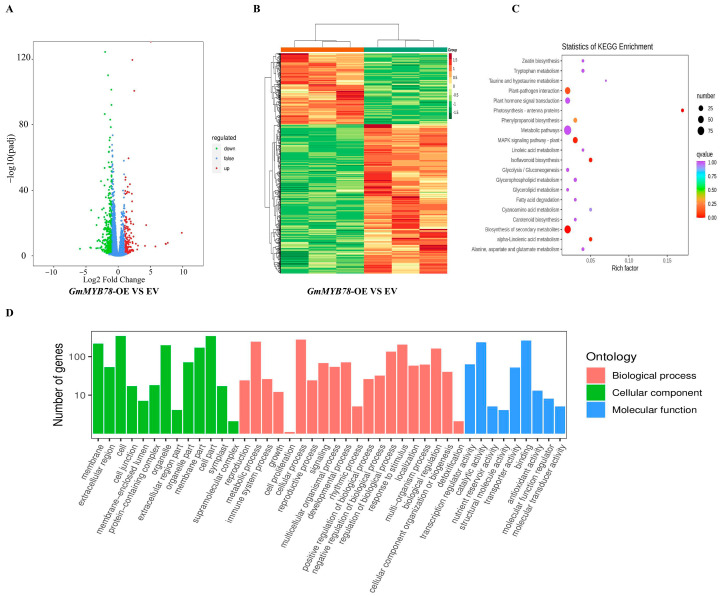
Transcriptomic analysis of gene expression profiles in response to *GmMYB78* overexpression. (**A**) Volcano plots of significantly differentially expressed genes in *GmMYB78*-OE vs. EV transgenic soybean hairy roots after the RNA-seq analysis. (**B**) Heat map of significantly differentially expressed genes between the EV and *GmMYB78*-OE transgenic soybean hairy roots, as determined using an RNA-seq analysis. Using a false discovery rate < 0.05 and a fold change ≥ 1 as the screening criteria, a total of 617 differentially expressed genes (DEGs) were identified. The scale bar indicates the fold changes (log2 values). (**C**) Gene annotation of KEGG classification of differentially expressed genes. The differentially expressed genes are mainly involved in multiple biological processes, including plant–pathogen interaction, and plant hormone signal transduction. (**D**) Gene Ontology functional classification of the differentially expressed genes. The differentially expressed genes were placed into the three main GO categories: biological process, cellular component, and molecular function.

**Figure 5 ijms-25-04247-f005:**
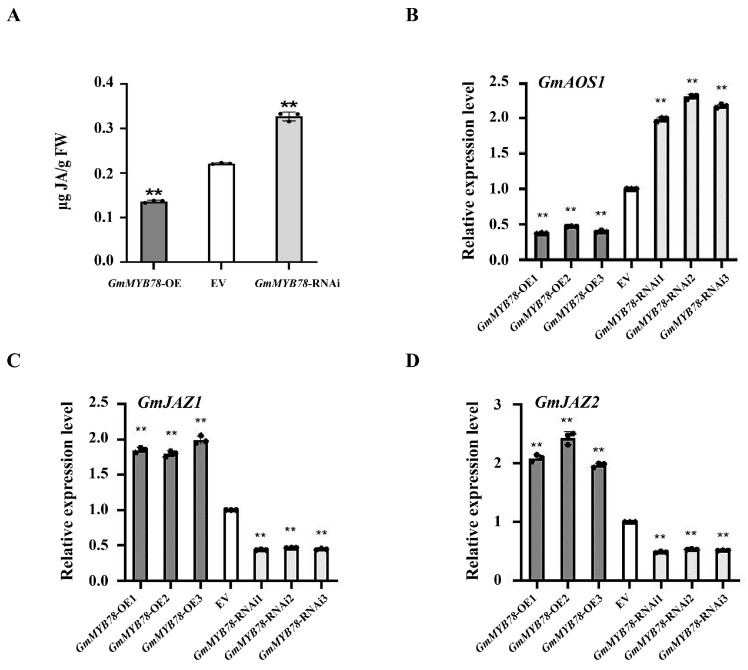
Investigation of the relationship between *GmMYB78* and the jasmonic acid (JA) pathway in soybean. (**A**) JA contents in *GmMYB78*-OE, *GmMYB78*-RNAi, and EV transgenic soybean hairy roots. (**B**–**D**) Relative transcript level of *GmAOS1* (**B**), *GmJAZ1* (**C**), or *GmJAZ2* (**D**) in *GmMYB78*-OE, *GmMYB78*-RNAi, and EV transgenic soybean hairy roots. The level of the control sample (EV lines) was set to unity. The experiment was statistically analyzed using Student’s *t*-test (** *p* < 0.01). Bars indicate the standard error of the mean.

**Figure 6 ijms-25-04247-f006:**
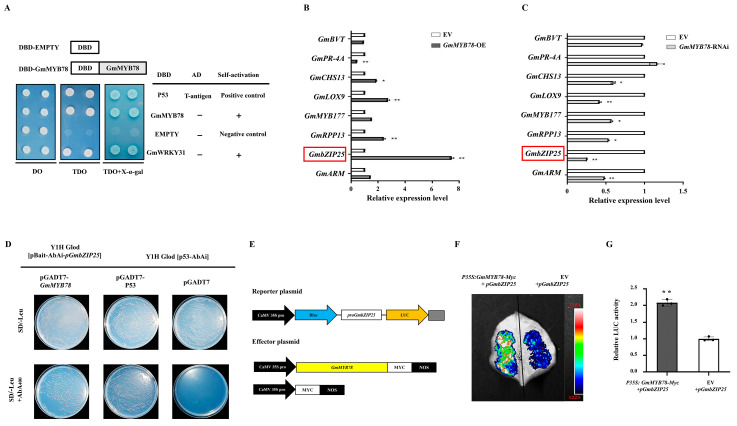
GmMYB78 directly regulates *GmbZIP25*. (**A**) The open reading frame (ORF) of *GmMYB78* was amplified into the pGBKT7 (GAL4 DBD) vector to generate the DBD-*GmMYB78* constructs. The yeast strain AH109 was transformed with pGBKT7-53 + pGADT7-T, pGBKT7-*GmMYB78*, pGBKT7-*GmWRKY3*,*1* and pGBKT7. The transformed cells were grown on synthetic dropout medium without tryptophan [DO], SD medium without Trp, histidine, and adenine [TDO], and SD medium without Trp, His, and Ade but with α-galactosidase [TDO + α-gal)] for 3 days at 30 °C. Transcriptional activation was monitored by the detection of yeast growth and performance of an α-Gal assay. (**B**,**C**) Relative expression of stress-related genes in the EV and (**B**) *GmMYB78*-OE and (**C**) *GmMYB78*-RNAi transgenic soybean hairy root. *GmPRR13* (Glyma.12G011700), *GmbZIP25* (Glyma.19G037900), *GmMYB177* (Glyma.14G210600), *GmPR-4A* (Glyma.20G225200), *GmLOX9* (XP_003524096.1), *GmCHS13* (Glyma.19G105100, *GmBVT* (Glyma.09G040400), and GmARM (Glyma.20G158500). The reference gene *GmEF1β* and *GmTUB4* were used as reference genes. The experiment was performed on three biological replicates, each with three technical replicates, and was statistically analyzed using Student’s *t*-test (* *p* < 0.05, ** *p* < 0.01). Bars indicate the standard error of the mean. *GmbZIP25* gene is labeled with the red boxes. (**D**) Verification of GmMYB78 regulating *GmbZIP25* in yeast. (**E**) The schematic diagram of reporter vector and effector vector. (**F**) Dual-luciferase assay in *N. benthamiana* leaves showing that GmMYB78 promotes the expression of *GmbZIP25*. Representative photographs are shown. (**G**) Detection of LUC/Rluc activity to verify that GmMYB78 promotes the expression of *GmbZIP25*. The combination of the reporter construct (*pGmbZIP25*: LUC) and EV was used as a control. The experiment was performed on three biological replicates, each with three technical replicates, and the results were statistically analyzed using Student’s *t*-test (* *p* < 0.05, ** *p* < 0.01). Bars indicate the standard deviation of the mean (*n* = 3).

**Figure 7 ijms-25-04247-f007:**
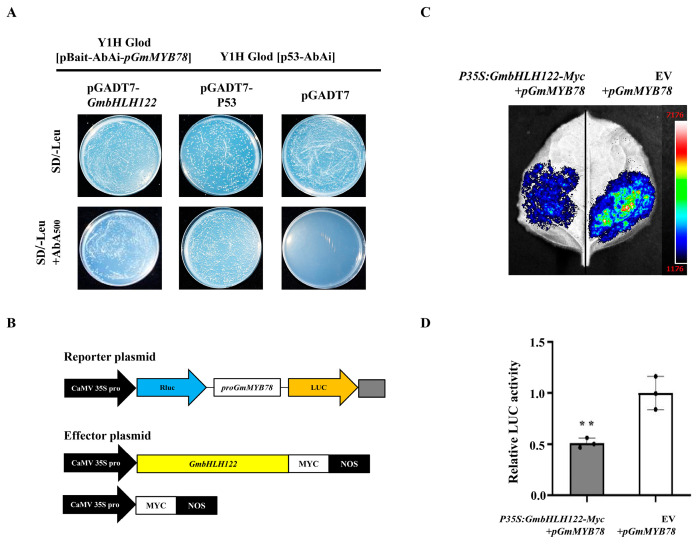
GmbLHL122 directly regulates *GmMYB78*. (**A**) Verification of GmbHLH122 regulating *GmMYB78* in yeast. (**B**) Schematic diagram of the reporter vector and effector vector. (**C**) Dual-luciferase assay in *N. benthamiana* leaves showing that GmbLHL122 promotes the expression of *GmMYB78*. Representative photographs are shown. (**D**) Detection of LUC/Rluc activity to verify that GmbLHL122 promotes the expression of *GmMYB78*. The combination of the reporter construct (*pGmMYB78*: LUC) and EV was used as a control. The experiment was performed on three biological replicates, each with three technical replicates, and the results were statistically analyzed using Student’s *t*-test (** *p* < 0.01). Bars indicate the standard deviation of the mean (*n* = 3).

**Figure 8 ijms-25-04247-f008:**
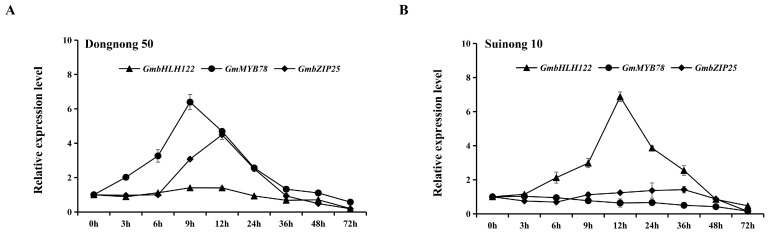
Temporal expression patterns of *GmbHLH122*, *GmMYB78*, and *GmbZIP25* in response to *P. sojae* infection in (**A**) the susceptible cultivar ‘Dongnong 50’ and (**B**) the resistant soybean cultivar ‘Suinong 10’. The expression was determined by qRT-PCR using *GmEF1β* and *GmTUB4* as the reference genes. Bars indicate the standard deviation of the mean (*n* = 3).

**Figure 9 ijms-25-04247-f009:**
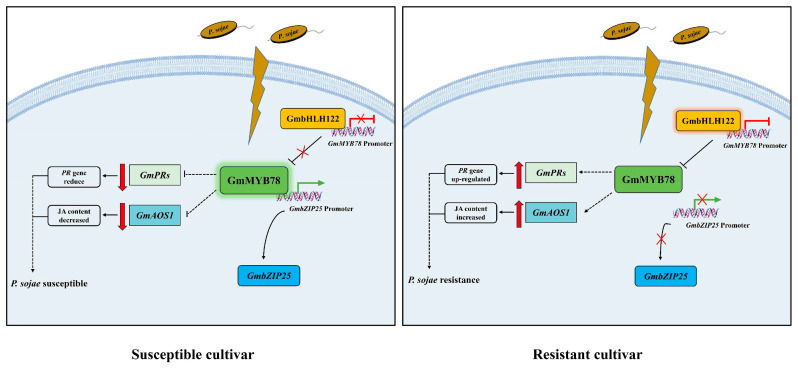
A proposed model for the molecular mechanism of the *GmMYB78* in soybeans responding to *P. sojae* infection. In the susceptible soybean cultivar ‘Dongnong 50’, *GmbHLH122* is not activated by infection, while *GmMYB78* is upregulated. The high levels of GmMYB78 suppresses the expression of *GmbZIP25* by binding to its promoter, and downregulates the expression of *GmPR* genes and JA-synthesis-pathway-related genes, and reduces JA content, thereby inhibiting the defense responses (**left**). In the resistant cultivar ‘Suinong 10’, *GmbHLH122* is activated by *P. sojae* infection. The high levels of GmbHLH122 inhibits *GmMYB78* transcription, which reduce its promotive effect on *GmbZIP25* expression, upregulates *GmPR* genes and JA synthesis pathway related genes, and promotes the accumulation of JA content, thereby enhancing the defense response to *P. sojae* (**right**).

## Data Availability

All data are represented in the article’s Appendix A.

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
