# Peer review of "The MYB Transcription Factor GmMYB78 Negatively Regulates Phytophthora sojae Resistance in Soybean"

_ijms, 2024, doi:10.3390/ijms25084247_

Round 1

Reviewer 1 Report

Comments and Suggestions for Authors

Due to the limitations of the hairy root system, in order to improve the scientificity of the data in this paper, it is suggested to provide relevant data of transgenic plants or to identify and distinguish the hairy roots more accurately.

Author Response

Thanks for the valuable suggestion from Reviewer 1. Our further experiments will involve transgenic plants. Following your valuable suggestions, we have provided the process description of soybean hairy root detection and phenotype identification and added the detailed method description in lines 656-662 and 667-670 in the section of Materials and methods in the revised manuscript.

Below are the detailed process description of soybean hairy root detection and phenotype identification.

1). Detection of soybean transgenic hairy roots: The transgenic hairy taproots will be used for P. sojae inoculation. The lateral roots grown on the taproots were tested to confirm the positive taproots. The transgenic hairy taproots with overexpression-target gene were tested via immunoblots, and RNAi transgenic hairy taproots were verified by QuickStix Kit for LibertyLink (bar) strips detection. The expression level of the overexpression and interference genes were also detected by RT-qPCR.

2). Assessment of soybean transgenic hairy roots in response to P. sojae infection: The healthy taproots with similar size were selected for P. sojae infection. When the hairy roots generated at the infection site grew approximately 3 weeks, which grew to around 7 cm and were detected as positive ones, the taproots were incubated with P. sojae zoospores (approximately 1 × 105 spores mL–1) in a mist chamber at 25 °C with 100% relative humidity for 72 h. The EV soybean hairy roots were used as controls. Disease symptoms on each taproot were observed after inoculation and photographed with a Nikon B7000 camera.

We have added the detailed description in the section of Material and Methods in the revised manuscript (lines 656-662 and 667-670).

Reviewer 2 Report

Comments and Suggestions for Authors

The manuscript by Gao et al selected a P. sojae induced GmMYB78 gene for function analysis by overexpression and RNAi in soybean hairy roots. The downstream and upstream genes were screened and verified by RNA-Seq, Y1H, and dual-luciferase assay. The MM is adequate and the experiments appear carefully performed. The data interpretation and the conclusions are appropriate. The findings are novel and have value to be published. I only have some line comments or suggestions as follows,

Line 15, RNA-Seq assay of GmAP2?

Line 19, JA biosynthesis-related genes were highly up-regulated à JA synthesis gene GmAOS1 was highly up-regulated

Line 106-108, add a citation here;RNA-seq analysis of GmAP2?

Line 124, Figure S2A à Figure S1A

Line 127, Figure S1B, add bootstrap value and distance scale. How to define an orthologue? More MYB proteins should be added, and papers related to genes clustered with GmMYB78 should be cited in introduction or discussion section.

Line 138, *P < 0.01?

Line 141, did not change significantly?

Line 155, A à C

Line 164, delete an

Line 170, or line 638, the soybean genotype used for transformation? And its sensitivity to P. sojae?

Line 173, Figure S2A, what sample in lane 4, between OE3 and EV?

Line 251, the full name and accession of GmAOS1

Line 279, Figure5A and 5B should be combined

Line 372, was down-regulated in GmAP2 transcriptome sequencing analysis à was down-regulated in GmAP2-OE hairy roots according to the transcriptome sequencing analysis

Line 375, ABA à AbA

Line 388, which part of the promoter was used in Y1H screen?

Line 390, Table S2 à Table S1

Author Response

  1. Line 15, RNA-Seq assay of GmAP2?

Response: Thanks for the valuable comments from Reviewer 2. For a clearer description, we have changed the ‘RNA-Seq assay of GmAP2’ to ‘transcriptome analysis of GmAP2-overexpressing transgenic hairy roots’ in the revised manuscript (lines 15-16).

  1. Line 19, JA biosynthesis-related genes were highly up-regulated à JA synthesis gene GmAOS1 was highly up-regulated

Response: Thanks for the valuable suggestions from Reviewer 2.  Following the suggestions, we have changed ‘JA biosynthesis-related genes were highly up-regulated’ to ‘JA synthesis gene GmAOS1 was highly up-regulated’ in the revised manuscript (lines 19-20).

  1. Line 106-108, add a citation here;RNA-seq analysis of GmAP2?

Response: Thanks for the elaborate suggestions from Reviewer 2. We have added the citation (Zhang et al., 2021) and changed the ‘RNA-Seq assay of GmAP2’ to ‘transcriptome analysis of GmAP2-overexpressing transgenic hairy roots’ in the revised manuscript (lines 108-110).

  1. Line 124, Figure S2A à Figure S1A

Response: Thanks for the valuable suggestions from Reviewer 2. We have changed Figure S2A to Figure S1A in the revised manuscript (line 125).

  1. Line 127, Figure S1B, add bootstrap value and distance scale. How to define an orthologue? More MYB proteins should be added, and papers related to genes clustered with GmMYB78 should be cited in introduction or discussion section.

Response: Thanks for the professional suggestions from Reviewer 2. Following the suggestions, we have added bootstrap value and distance scale in the Figure S1B in the revised manuscript. All the genes contained in Figure S1B can only be defined as homologous genes, so we changed orthologue to homologous genes in the revised manuscript (line 127). Furthermore, we also added more MYB proteins to Figure S1B and cited papers related to genes clustered with GmMYB78 in the discussion (lines 128-143 and 470-472).

  1. Line 138, *P < 0.01?

Response: Thanks for the valuable suggestions from Reviewer 2. We have changed (*P < 0.01) to (**P < 0.01) in the revised manuscript (line 149).

  1. Line 141, did not change significantly?

Response: Thanks for the professional suggestions from Reviewer 2. We have changed ‘did not change significantly’ to ‘was rapidly down-regulated at 6 h post inoculation (hpi)’ in the revised manuscript (lines 152-153).

  1. Line 155, A à C

Response: Thanks for the elaborate suggestions from Reviewer 2. We have changed A to C in the revised manuscript (line 166).

  1. Line 164, delete an

Response: Thanks for the elaborate suggestions from Reviewer 2. We have deleted ‘an’ in the revised manuscript (line 172).

  1. Line 170, or line 638, the soybean genotype used for transformation? And its sensitivity to P. sojae?

Response: Thanks for the valuable suggestions from Reviewer 2. We used P. sojae susceptible soybean cultivar ‘Dongnong 50’ for transformation, and we have added the description in revised manuscript (lines 181 and 649-650).

  1. Line 173, Figure S2A, what sample in lane 4, between OE3 and EV?

Response: Thanks for the valuable comments from Reviewer 2. The lane 4 is the repeat of OE3, which is the same as lane 3. In order to ensure the integrity of immunoblotting, we did not remove it.

  1. Line 251, the full name and accession of GmAOS1

Response: Thanks for the elaborate comments from Reviewer 2. We have added the full name (Allene Oxide Synthase 1) and accession (GenBank accession no. NP_001236432.1) of GmAOS1 in the revised manuscript (lines 259-260).

  1. Line 279, Figure5A and 5B should be combined

Response: Thanks for the professional comments from Reviewer 2. Following the suggestions, we have combined Figure 5A and 5B in the revised manuscript (Figure 5A).

  1. Line 372, was down-regulated in GmAP2 transcriptome sequencing analysis à was down-regulated in GmAP2-OE hairy roots according to the transcriptome sequencing analysis

Response: Thanks for the valuable suggestions from Reviewer 2. Following the suggestions, we have changed ‘was down-regulated in GmAP2 transcriptome sequencing analysis’ to ‘was down-regulated in GmAP2-OE hairy roots according to the transcriptome sequencing analysis’ in the revised manuscript (lines 378-379).

  1. Line 375, ABA à AbA

Response: Thanks for the valuable comments from Reviewer 2. We have changed ABA to AbA in the revised manuscript (line 381).

  1. Line 388, which part of the promoter was used in Y1H screen?

Response: Thanks for the professional suggestions from Reviewer 2. We used the 113bp-670bp promoter of GmMYB78 in Y1H screening, and we have added the description in revised manuscript (lines 396-397).

  1. Line 390, Table S2 à Table S1

Response: Thanks for the valuable comments from Reviewer 2. We have changed Table S2 to Table S1 in the revised manuscript (line 398).

Reviewer 3 Report

Comments and Suggestions for Authors

This manuscript investigates the role of GmMYB78 in soybean resistance to Phytophthora sojae. Through experiments, it is revealed that GmMYB78 enhances sensitivity to the pathogen. However, the mechanism behind this phenomenon, particularly regarding JA signaling pathway inhibition, requires further clarification. The study also identifies a cascade pathway involving GmbHLH122-GmMYB78-GmbZIP25. Despite minor errors in spelling, grammar, and clarity, the manuscript provides valuable insights into soybean defense mechanisms against Phytophthora sojae infection.

Minor comments

Abstract:

Line 13: Phythophthora should be Phytophthora

Line 14: Change GmAP2 enhanced sensitivity to GmAP2 enhances sensitivity.

Line 15: Clarify the meaning of down-regulated in RNA-Seq assay of GmAP2.

Line 17: Rephrase but the opposite is true, indicating that for clarity.

Line 26: Clarify the grammatical correctness of sensitivity of soybean to P. sojae via inhibiting JA signaling pathway.

Line 27: Clarify or rephrase GmbHLH122-GmMYB78-GmbZIP25 cascade pathway for better understanding.

Introduction:

Line 119: Correct GeneBank to GenBank.

Line 150: Clarify only observed GmMYB78-GFP and nuclear marker for grammatical correctness.

Line 155: Change representation to representations.

Results:

Line 193: Provide a definition for PSPEL1 for clarity.

Line 194: Avoid repeating GmEF1β and GmTUB4 unnecessarily.

Line 219: Omit unnecessary repetition of GmEF1β and GmTUB4.

Line 235: Replace hormone with jasmonic acid for clarity.

Line 250: Replace hormone synthesis pathway with JA synthesis pathway for clarity.

Line 269: Clarify the meaning of soybean to P. sojae.

Author Response

Abstract:

  1. Line 13: Phythophthora should be Phytophthora

Response: Thanks for the valuable comments from Reviewer 3.  We have changed Phythophthora to Phytophthora in the revised manuscript (line 13).

  1. Line 14: Change GmAP2 enhanced sensitivity to GmAP2 enhances sensitivity.

Response: Thanks for the professional suggestions from Reviewer 3. Following the suggestions, we have changed ‘GmAP2 enhanced sensitivity’ to ‘GmAP2 enhances sensitivity’ in the revised manuscript (lines 14-15).

  1. Line 15: Clarify the meaning of down-regulated in RNA-Seq assay of GmAP2.

Response: Thanks for the valuable comments from Reviewer 3.  For a clearer description, we have changed the ‘down-regulated in RNA-Seq assay of GmAP2’ to ‘down-regulated in transcriptome analysis of GmAP2-overexpressing transgenic hairy roots’ in the revised manuscript (lines 15-16).

  1. Line 17: Rephrase but the opposite is true, indicating that for clarity.

Response: Thanks for the professional suggestions from Reviewer 3. Following the suggestions, we have rephrased ‘but the opposite is true’ as ‘while silencing GmMYB78 enhances resistance to P. sojae’ for clarity in the revised manuscript (line 18).

  1. Line 26: Clarify the grammatical correctness of sensitivity of soybean to P. sojae via inhibiting JA signaling pathway.

Response: Thanks for the valuable comments from Reviewer 3. Following the suggestions, we have modified it to ‘In conclusion, our data reveal that GmMYB78 triggers soybean sensitivity to P. sojae by inhibiting the JA signaling pathway and the expression of pathogenesis-related genes or through the effects of GmbHLH122-GmMYB78-GmbZIP25 cascade pathway.’ to improve the clarity of the expression. We have added the description in revised manuscript (lines 27-29).

  1. Line 27: Clarify or rephrase GmbHLH122-GmMYB78-GmbZIP25 cascade pathway for better understanding.

Response: Thanks for the professional suggestions from Reviewer 3. Following the suggestions, we have clarified GmbHLH122-GmMYB78-GmbZIP25 cascade pathway for better understanding, and added the description in revised manuscript (lines 23-29) as following: Additionally, we screened and identified the upstream regulator GmbHLH122 and downstream target gene GmbZIP25 of GmMYB78.  GmbHLH122 was highly induced by P. sojae and could inhibit GmMYB78 expression in resistant soybean, and GmMYB78 was highly expressed to activate downstream target gene GmbZIP25 transcription in susceptible soybean.

Introduction:

  1. Line 119: Correct GeneBank to GenBank.

Response: Thanks for the elaborate comments from Reviewer 3. The ‘GeneBank’ was corrected to ‘GenBank’ in the revised manuscript (line 111).

  1. Line 150: Clarify only observed GmMYB78-GFP and nuclear marker for grammatical correctness.

Response: Thanks for the professional suggestions from Reviewer 3. We have changed ‘only observed GmMYB78-GFP and nuclear marker’ to ‘only GmMYB78-GFP and nuclear marker were observed’ in the revised manuscript (lines 162-163).

  1. Line 155: Change representation to representations.

Response: Thanks for the elaborate comments from Reviewer 3. The representation was corrected to representations in the revised manuscript (lines 174-175).

Results:

  1. Line 193: Provide a definition for PSPEL1 for clarity.

Response: Thanks for the valuable comments from Reviewer 3.  We have changed PSPEL1 to PSEL1 and provided the definition for PSEL1 (P. sojae elicitin gene 1) in the revised manuscript (lines 204-205).

  1. Line 194: Avoid repeating GmEF1β and GmTUB4 unnecessarily.

Response: Thanks for the valuable suggestions from Reviewer 3. We have deleted GmEF1β and GmTUB4 in the revised manuscript (lines 205).

  1. Line 219: Omit unnecessary repetition of GmEF1β and GmTUB4.

Response: Thanks for the professional suggestions from Reviewer 3. Following the suggestions, we have omitted unnecessary repetition of GmEF1β and GmTUB4 in the revised manuscript (line 226)。

  1. Line 235: Replace hormone with jasmonic acid for clarity.

Response: Thanks for the professional suggestions from Reviewer 3. Following the suggestions, we have replaced ‘hormone’ with ‘jasmonic acid’ for clarity in the revised manuscript (line 242).

  1. Line 250: Replace hormone synthesis pathway with JA synthesis pathway for clarity.

Response: Thanks for the professional comments from Reviewer 3.  Following the suggestions, we have replaced ‘hormone synthesis pathway’ with ‘JA synthesis pathway’ for clarity in the revised manuscript (line 258).

  1. Line 269: Clarify the meaning of soybean to P. sojae.

Response: Thanks for the valuable comments from Reviewer 3. We have changed ‘soybean to P. sojae’ to ‘soybean resistance to P. sojae’ in the revised manuscript (line 277).
